# Nationwide implementation of the non-invasive prenatal test: Evaluation of a blended learning program for counselors

**Linda Martin** [1]*, **Janneke T. Gitsels-van der Wal** [1], **Caroline J. Bax** [2], **Mijntje J. Pieters** [3,4], **Jacqueline C. I. Y. Reijerink-Verheij** [5], **Robert-Jan Galjaard** [6], **Lidewij Henneman** [7], **Dutch NIPT Consortium** [¶]

1 Amsterdam UMC, Vrije Universiteit Amsterdam, Midwifery Science, AVAG, Amsterdam, The Netherlands, 2 Amsterdam UMC, University of Amsterdam, Dept of Obstetrics, Amsterdam, The Netherlands, 3 Foundation Prenatal Screening Southeast region of the Netherlands, Maastricht, The Netherlands, on behalf of the Regional Centres for Prenatal Screening, The Netherlands, 4 Department of Obstetrics and Gynecology, GROW School of Oncology and Developmental Biology, Maastricht University Medical Centre, Maastricht, The Netherlands, 5 Foundation Prenatal Screening Southwest region of the Netherlands, Rotterdam, on behalf of the Regional Centres for Prenatal Screening, The Netherlands, 6 Erasmus MC, Dept. of Clinical Genetics, Rotterdam, The Netherlands, 7 Amsterdam UMC, Vrije Universiteit Amsterdam, Dept of Clinical Genetics and Amsterdam Reproduction and Development research institute, Amsterdam, The Netherlands

☯ These authors contributed equally to this work.
¶ Membership of the Dutch NIPT Consortium is listed in the S3 Appendix.
* linda.martin@inholland.nl

**Data Availability Statement:** The data underlying the results presented in the study are available

## Abstract

This study assesses the results of a mandatory blended learning-program for counselors (e.g. midwives, sonographers, obstetricians) guiding national implementation of the Non-Invasive Prenatal Test (NIPT). We assessed counselors' 1) knowledge about prenatal aneuploidy screening, 2) factors associated with their knowledge (e.g. counselors' characteristics, attitudes towards NIPT), and 3) counselors' attitudes regarding the blended learning. A cross-sectional online pretest-posttest implementation survey was sent to all 2,813 Dutch prenatal counselors. Multivariate linear regression analyses were performed to identify associations between counselors' knowledge and e.g. their professional backgrounds, work experience and attitudes towards NIPT. At T0 and T1 1,635 and 913 counselors completed the survey, respectively. Overall results show an increased mean number of correct answered knowledge questions; 23/35 (66%) items at T0 and 28/37 (76%) items at T1. Knowledge gaps on highly specific topics remained. Work experience and secondary care work-setting were positively associated with a higher level of knowledge. Most counselors (74%) showed positive attitudes towards the blended learning program. The mandatory blended learning, along with learning by experience through implementation of NIPT, has facilitated an increase in counselors' knowledge and was well received. New implementations in healthcare may benefit from requiring blended learning for healthcare providers, especially if tailored to professionals' learning goals.

from Jens Henrichs, PhD, Amsterdam UMC-VUMC: j.henrichs@amsterdammumc.nl.

**Funding:** This study was partly supported by a grant from the Netherlands Organization for Health Research and Development (ZonMw, No. 543002001). With regards to our Financial Disclosure we state: "This study was partly supported by a grant from the Netherlands Organization for Health Research and Development; grant applicant LH (ZonMw, No. 543002001, https://www.zonmw.nl/nl/). The funder had no role in study design, data collection and analysis, decision to publish, or preparation of the manuscript.

**Competing interests:** The authors have declared that no competing interests exist.

## Introduction

Non-Invasive Prenatal Testing (NIPT) is increasingly available worldwide, as the field of prenatal screening for fetal anomalies is rapidly implementing new genetic technologies [1]. NIPT is a maternal blood test using cell-free DNA, mostly targeted at the common aneuploidies Down-, Edwards- and Patau syndrome. In most countries, NIPT is readily commercially available, although it has increasingly been offered as an aneuploidy screening test, within nationwide healthcare programs, in which mostly primary maternity care providers offer accompanying pre-test counseling [1–3]. Given this context, obstetric care providers have had to adapt by updating their knowledge about NIPT rapidly. Such knowledge, e.g. understanding the advantages and limitations of NIPT compared to other prenatal tests and knowing how to interpret the results, is essential to allow an adequate pregnant women's informed choice regarding prenatal screening [4–7].

Given the importance of counselors' up-to-date knowledge about NIPT, education of counselors has been an active topic for policymakers and educators. This has resulted in the development of programs aimed at improving counselors' knowledge and communicational skills [5, 8]. In a public healthcare system, counselors' education on NIPT typically involves a diverse palette of educational activities, so-called blended learning. Blended learning includes, for instance, face-to-face training, e-learning, assessments, and web-based information such as guidelines and Option Grid decision aids [9–12]. One of the challenges of educational programs is that counselors' professional backgrounds (e.g. obstetrician, midwife, sonographers) and therefore professional knowledge differs significantly [9]. Furthermore, despite efforts to educate counselors properly, certain aspects such as the origin of cell-free fetal DNA used for NIPT and discordant/false-positive rates of NIPT seem difficult to understand [6, 9, 13]. The resulting knowledge gaps are problematic since insight into NIPT's test characteristics is fundamental for understanding the clinical implications of the test results and the fact that NIPT is not diagnostic, diagnostic options and thereby for providing adequate counseling that supports informed decisions [4, 6, 14, 15].

Alongside these difficulties in education, providers offering counseling for prenatal aneuploidy screening report additional challenges such as biased education by commercial test providers, lack of nationwide consensus about the educational content, and trouble maintaining up-to-date knowledge about NIPT and its target anomalies [2, 5]. The challenges also result from the rapidly widening scope of NIPT, the diversity of analyses used by different laboratories, and the resulting lack of sustainable, reliable knowledge about NIPT that accompanies the evolving technical accuracy of NIPT [16, 17]. Furthermore, relatively little is known about efficient and effective educational programs accompanying implementation of NIPT, to address the educational difficulties counselors face. Finally, there is limited knowledge about the impact of counselors' attitudes regarding NIPT on outcomes of the educational programs [18]. Such information is essential to learn more about how educational programs can be developed, implemented and improved to provide efficient and effective strategies for counselor education.

In the Netherlands, national studies (TRIDENT-1, TRIDENT-2) were launched to evaluate the organizational implementation of NIPT, test characteristics and outcomes, women's perspectives as well as the counselors' perspectives, and the educational requirements. In 2017, alongside the transition of NIPT from a second-tier test, only available for women with a high risk pregnancy based on the First trimester Combined Test results or medical indication, to a first-tier screening test for all pregnant women, the Dutch government set new educational requirements for healthcare professionals who provide counseling for prenatal aneuploidy screening [19, 20] (S1 Appendix). In addition to existing requirements for continuing

| 1. February-March 2017 Pre-implementation knowledge assessment (T0) | 2. May 2017 E-learning including a knowledge examination | 3. March 2017 Face-to-face seminars | 4. March 2017 Counseling guideline and factsheets for counselors | 5. April 2017 Nationwide NIPT Implementation, start TRIDENT-2 | 6. May 2017 Websites with help-desk for both counselors and clients | 7. November 2017 - December 2018 Counseling skills training | 8. November 2017 - January 2018 Post-implementation knowledge assessment (T1) |
|---|---|---|---|---|---|---|---|

**Fig 1. Overview of the blended learning for counselors and timing of knowledge assessment (T0 and T1) NIPT, Non-Invasive Prenatal Testing.**

education, counselors were obliged to participate in a blended learning program about NIPT to improve their knowledge and skills on this topic [11, 21] (Fig 1). This program addressed the first three of the four levels of Miller's Pyramid of Professional Competence (knows, knows how, shows how and does) [22]. Our study aimed to assess the results of this mandatory blended learning program guiding national implementation of NIPT by evaluating: 1) counselors' knowledge about prenatal aneuploidy screening, 2) factors associated with knowledge (e.g. counselor's characteristics, attitudes towards NIPT), and 3) attitudes regarding the blended learning program. Although the blended learning program addressed the first three levels of Miller's Pyramid of Professional Competence, here we only evaluated the effects of the blended learning program on the first level 'knows'.

## Materials and methods

The current study was part of the TRIDENT-2 study to evaluate the introduction of NIPT as first-tier screening in a governmentally-funded prenatal aneuploidy screening program [9, 19, 23–25].

### Study design and procedures

A pre- and post-implementation survey study design was developed using an online knowledge questionnaire distributed to all registered prenatal counselors. The first assessment (T0) took place February-March 2017, before the start of the NIPT blended learning program and implementation of NIPT as first-tier aneuploidy screening test. To facilitate tailoring of the blended learning activities, we provided Regional Centers for prenatal screening (RCs) with the overall anonymous results of pre-implementation knowledge scores of the counselors. The second measurement (T1) took place approximately nine months later, after implementing the blended learning program and first-tier NIPT (November 2017-January 2018). Study participation was voluntary for completion of both questionnaires; no incentives were given.

### Measures

We asked for counselors' characteristics (age, profession, work experience, region), counseling education and number of counseling sessions a month. Furthermore, at both T0 and T1, counselors' attitudes towards NIPT as first-tier aneuploidy screening test were measured using one, closed-ended question "What do you think about offering NIPT as a first-tier screening test?" (answer options: 'good', 'not good', 'neutral').

### Knowledge questionnaire

The knowledge questionnaire was developed based on questionnaires used in previous Dutch studies [6, 13, 26–28] and international literature [4, 6, 29] (S2 Appendix). Development was done by a multidisciplinary team, consisting of representatives from midwifery, gynecology, psychology, clinical genetics and health sciences. After T0, the 38- item questionnaire was adjusted based on feedback from participants, and policy officers from the Centre for Population Screening at the National Institute for Public Health and the Environment (RIVM-CvB)

and the RCs; unclear questions (N = 3) were excluded, and missing knowledge domains were added (N = 2). The resulting 35-item knowledge questionnaire at T0 and a 37-item knowledge questionnaire at T1 required the answers 'true', 'false' or 'I do not know', and consisted of seven themes/domains: *1) Prenatal aneuploidy screening program, 2) Test characteristics of NIPT, 3) NIPT versus FCT, 4) Additional findings from NIPT and FCT, 5) In- and exclusion-criteria for NIPT and FCT, 6) Follow-up tests, and 7) Fetal structural Anomaly Screening.*

### Attitude regarding the blended learning education

At T1, counselors' attitudes towards the mandatory blended learning were measured by one question 'What do you think about the mandatory blended learning education on providing the knowledge and counseling skills you have to complete to stay certified as a counselor?' to be answered as 'good', 'not good', or 'other' including space to explain the given answer.

### Data collection

The questionnaire was distributed to all 2813 certified Dutch counselors [30]. Counselors received an email from their RC with a request to participate. The email contained a link to the online questionnaire and information about the research, such as the study's purpose, voluntary participation and privacy guarantee. With this method, we aimed to prevent as much selection and information bias as possible. Two and four weeks after the questionnaire's distribution, reminders were sent for both T0 and T1 requesting the completion of the questionnaire.

### Analyses

Descriptive statistics (N, %, mean (M), or standard deviation (SD) where appropriate) were used to describe the background characteristics of participating counselors, including attitudes towards NIPT. We compared characteristics of the respondents of T0 versus T1, and to available characteristics of the Dutch counselor population to examine the representativeness and comparability of different professions (e.g. midwife, gynecologist, sonographer) and regions. Counselors' professional background were dichotomized according to their work setting into 'primary care' (e.g. midwives and sonographers) and 'secondary/tertiary care' (e.g. gynecologists, medical doctors in resident and nurses).

**Counselors' pre- and post-blended learning NIPT knowledge.** Before the analyses, the answers to the knowledge questions were dichotomized into 'correct' or 'not correct' categories; the answer 'I do not know' was coded as 'not correct'. Subsequently, numbers and percentages of correct answers were calculated per question, per knowledge theme and as an average of the total amount of correctly answered questions. Analyses were subsequently repeated for counselors connected to each of the RCs. If participants had answered at least 70.0% of the questions correctly, the same cutoff as used in the national e-learning and assessment developed by the RIVM-CvB (https://www.pns.nl/professionals/nipt-seo/scholing-counselors/e-learning), their knowledge was considered sufficient.

For analyzing the potential influences of the NIPT blended learning program and nine months of experience with NIPT as a first-tier test on counselors' knowledge, frequencies of correctly answered questions were calculated per knowledge item. In addition, the sum of correctly answered items for T0 and T1 were calculated separately for the whole group and each RC.

We used multivariable, linear regression analyses to identify differences between results of the blended learning education of counselors' knowledge at T0 and T1. Separate models were analyzed for T0 and T1, because we did not want to lean on the assumption that the same

participants were included at both assessments. Backward selection was performed on the initial multivariable model for the sequential removal of variables (profession, work setting, work experience, number of counseling sessions a month, and attitude towards NIPT). In each step, the variable with the highest p-value was removed until the model contained only statistically significant variables (two-sided p < 0.05). One sample proportion tests were performed to identify significant differences between the results of T0 and T1.

**Attitudes regarding the blended learning.** Attitudes regarding the mandatory blended learning education for counselors were described by applying descriptive statistics (N, %, M, SD where appropriate). All analyses were carried out using SPSS 24.0, and a significance level of p≤0.05 was maintained.

### Ethical considerations

The study protocol was approved by the Medical Ethical Committee of the VU University Medical Center, Amsterdam, the Netherlands (no. 2017.165). We collected online written consent using a 'yes' or 'no' button from all participating counselors, before they could contribute in the online survey.

## Results

Table 1 (Counselors' characteristics pre- (T0) and post (T1) NIPT implementation and participation in the blended learning program) shows the background characteristics and participation of respondents. At T0, 1,635 counselors completed the knowledge questionnaire (response 1,635/2813 = 58%) and at T1, 913 counselors did so (response 913/2,813 = 32%). At both measurements, most counselors worked in primary care (83% and 80%, respectively); the majority worked as a midwife. At T1 significantly more counselors (86%) showed a positive attitude towards NIPT as a first-tier aneuploidy screening compared to T0 (75%). At T0, self-reported mean years of work experience as a counselor was M = 9.7 (SD = 5.6) years and at T1, M = 8.6 (SD = 5.6) years. At T1, the number of counseling consultations was M = 9.2 (SD = 7.6) per month.

### Counselors' pre- and post- blended learning NIPT knowledge

Table 2 (Counselors' correctly answered knowledge questions at T0 (N = 1,635) and T1 (N = 913) shows counselors' pre- and post-blended learning of NIPT as percentage of counselors who correctly answered the items in various knowledge themes. Analyses of sum scores of the respectively 35 and 37 items at T0 and T1 showed that counselors answered significantly more items correctly at T1 compared to T0; overall mean score of M = 28 (76%; SD = 3.0) correct answers versus M = 23 (66%; SD = 3.8), respectively. The overall mean scores per RC are shown in Table 3 (Overall mean scores of counselors' knowledge represented per Prenatal Screening Region). Mean scores per RC at T0 varied between 22.5–24.0 (64%-69%) correctly answered questions, and at T1 between 28.0–29.6 (76%-80%) correctly answered questions. There were no significant difference in knowledge scores between RCs.

Considering the scores on the separate knowledge themes, at both T0 and T1, all items of the themes *NIPT versus FCT* and *Fetal structural Anomaly Screening (FAS)* were correctly answered by more than 70% of the participating counselors. At T0 all six items of the theme *In- and exclusion-criteria NIPT and FCT* were answered correctly by 21% - 55% of the counselors. At T1, after the blended learning program, at least 70% of the counselors answered four of these 6 questions correctly. The remaining two items of this theme (on monochorionic pregnancies) were answered correctly by 53% and 63% of the. Half of the items on the anomalies

**Table 1. Counselors' characteristics pre- (T0) and post (T1) NIPT implementation and participation in the blended learning program.**

| Characteristics | T0 | T1 | Dutch counselor population |
|---|---|---|---|
| | N = 1,635 | N = 913 | N = 2,813[5] |
| **Age** mean (SD) in years | 37.0 (10.9) | 38.0 (11.1) | - |
| **Experience** (in years), n (%) | | | |
| ≤5 years | 395 (24.2) | 268 (29.4) | - |
| 6–10 years | 487(29.8) | 406 (44.5) | |
| ≥11 years | 753 (46.0) | 239 (26.1) | |
| **Profession**[1], n (%) | | | |
| Midwife primary care | 1,358 (83.1) | 727 (79.6) | 2,125 (79.1) |
| Midwife secondary care | 110 (6.7) | 41 (4.5) | 79 (3.0) |
| Gynecologist | 91 (5.6) | 41 (4.5) | 40 (1.5) |
| Sonographer[2] | 244 (14.9) | 52 (5.7) | 441 (16.4) |
| Other[3] | 80 (4.9) | 47 (5.1) | |
| Regional Prenatal Screening Center[4], n (%) | | | |
| Region 1 | 323 (19.7) | 216 (23.7) | |
| Region 2 | 255 (15.6) | 138 (15.1) | |
| Region 3 | 190 (11.6) | 78 (8.5) | |
| Region 4 | 147 (9.0) | 80 (8.8) | |
| Region 5 | 302 (18.5) | 185 (20.3) | |
| Region 6 | 234 (14.3) | 156 (17.1) | |
| Region 7 | 207 (12.7) | 87 (9.5) | |
| **Attitude toward NIPT**, n (%) | | | |
| Positive | 1,228 (74.7) | 724 (85.6) | |
| Negative | 117 (7.1) | 35 (4.1) | |
| Neutral | 299 (18.2) | 87 (10.3) | |
| **Participation in blended learning activities** | | | |
| **E-learning completed**, n (%) | | | |
| Yes | Not | 909 (99.6) | |
| No | applicable | 4 (0.4) | |
| **Attended face-to-face seminar,** n (%) | | | |
| Yes | Not | 910 (99.7) | - |
| No | applicable | 3 (0.3) | |

Valid percentages are shown

[1] Participants might have had more than one profession and answered accordingly; therefore, numbers add up to more than N = 1,635/N = 913

[2] Most sonographers were also midwives

[3] Gynecologist in residence, nurse, reproductive physician

[4] Data of the two Regional Prenatal Screening Centers in Amsterdam are merged

[5] Annual Report of all national Regional Prenatal Screening Centers 2018.

screened for (*Trisomy 13, 18 and 21*) were answered correctly by less than 70% of the counselors at T1.

Multivariate linear regression analyses showed that both at T0 and T1, more years of work-experience and a secondary-care work setting were associated with significantly more correctly answered knowledge questions (T0: $b = 0.068$, $p < 0.0001$ and $b = 0.003$, $p < 0.001$; T1: $b = 0.056$; $p = 0.002$ and $b = 0.002$; $p < 0.0001$, respectively).

**Table 2. Counselors' correctly answered knowledge questions at T0 (N = 1,635) and T1 (N = 913).**

| Knowledge Theme | Question | Answer | T0: N (%) | T1: N (%) |
|---|---|---|---|---|
| *Prenatal anomaly screening program* | The pregnant woman is free to decide whether or not to have a fetal anomaly scan. | True | 1,634 (99.9) | 911 (99.8) |
| | If a pregnant woman is NOT considering terminating the pregnancy, then it is NOT necessary to discuss the prenatal, anomaly screening options. | False | 1,625 (99.4) | 910 (99.7) |
| | The pregnant woman has the right to decide whether or not to receive information about prenatal screening options. | True | 1,623 (99.3) | 907 (99.3) |
| | As a counselor, it is important to advice the pregnant woman whether or not to opt for prenatal anomaly screening, because a counselor has a better insight into the risks than the pregnant woman herself. | False | 1,579 (96.6) | 887 (97.2) |
| | If a woman chooses NIPT, she can also have a nuchal fold measurement taken as part of the screening program. | False | 650 (39.8) | 823 (90.1) |
| | Women who had previously been pregnant with a child with trisomy 21, 18, 13 only receive counseling in a prenatal diagnosis center for screening for trisomy 21, 18 and 13. | True | 417 (25.5) | 627 (68.7) |
| | If NIPT finds signs of cancer in a pregnant woman, this is always reported to her even if she did not opt for disclosure of incidental findings. | True | not applicable | 622 (68.1) |
| *Trisomy 13, 18, 21* | Most infants with Edward's syndrome die before birth or shortly after birth. | True | 1,602 (98.0) | 900 (98.6) |
| | The most common form of Down syndrome is hereditary. | False | 1,515 (92.7) | 850 (93.1) |
| | Approximately half of all infants born with Down syndrome have a heart defect. | True | 1,361 (83.2) | 815 (89.3) |
| | Delays in the development of motor skills in an infant with Down syndrome does NOT affect other areas of development. | False | 1,117 (68.3) | 614 (67.3) |
| | Most babies with Down syndrome are born to women under the age of 36. | True | 1,019 (62.3) | 633 (69.3) |
| | 5–10% of infants with trisomy 13 survive beyond the first year of life. | True | 693 (42.4) | 449 (49.2) |
| *Test characteristics of NIPT* | NIPT determines whether the fetus is healthy. | False | 1,631 (99.8) | 910 (99.7) |
| | If the result of NIPT is: "negative for trisomy 21,18 and 13", then there is NO chance that the baby has trisomy 21,18 and 13. | False | 1,313 (80.3) | 753 (82.5) |
| | If NIPT gives a positive result for trisomy 21 in the initial screening, there is an average 25% chance that the infant does NOT have trisomy 21. | True | 921 (56.3) | 655 (71.7) |
| | The cell-free fetal DNA in maternal blood used for NIPT comes from the placenta. | True | 899 (55.0) | 827 (90.6) |
| | A failed result in NIPT (no result) occurs in 2 out of 100 tests. | True | 867 (53.0) | 657 (72.0) |
| | If NIPT gives a positive result for trisomy 13 in the initial screening (Patau syndrome), there is an average of a 75% probability that the infant does NOT have trisomy 13. | True | 174 (10.6) | 428 (46.9) |
| *NIPT versus FCT* | NIPT has a higher sensitivity than the first trimester combined test. | True | 1,463 (89.5) | 880 (96.4) |
| | When using NIPT as the initial screening test, fewer pregnant women are sent for follow-up testing than after the first trimester combined test. | True | 1,363 (83.4) | 830 (90.9) |
| | For initial screening, the personal cost for the first trimester combined test and NIPT are approximately the same as of 1-4-2017. | True | 1,224 (74.9) | 882 (96.6) |
| *Additional findings NIPT and FCT* | Additional findings (other than trisomy 21, 18 or 13) can result from the first trimester combined test. | True | 1,119 (68.4) | 505 (55.3) |
| | For initial screening with NIPT, the pregnant woman can choose whether she wants to hear additional findings. | True | 840 (51.4) | 901 (98.7) |
| | If the pregnant woman DOES want to know additional findings, chromosomes other than chromosomes 21, 18 and 13 can also be examined. NIPT results can include more than trisomy 21, 18 and 13. | True | 766 (46.9) | 856 (93.8) |

*(Continued)*

**Table 2.** (Continued)

| Knowledge Theme | Question | Answer | T0: N (%) | T1: N (%) |
|---|---|---|---|---|
| | As an incidental finding, abnormalities of the placenta can also be detected by NIPT. | True | not applicable | 741 (81.2) |
| *Inclusion and exclusion criteria for NIPT and FCT* | The first trimester combined test CANNOT be performed if a woman is pregnant with monochorionic twins. | False | 898 (54.9) | 640 (70.1) |
| | A thickened nuchal fold (≥3.5 mm) is NOT an indication for NIPT. | True | 814 (49.8) | 794 (76.0) |
| | NIPT CAN be performed if a woman is pregnant with monochorionic twins. | True | 741 (45.3) | 481 (52.7) |
| | Monochorionic twin pregnancies are monozygotic. | True | 489 (29.9) | 577 (63.2) |
| | If a pregnant woman has a chromosomal abnormality herself, she may NOT eligible for NIPT. | True | 384 (23.5) | 649 (71.1) |
| | If a pregnant woman is 17 years old, she CAN be screened by NIPT. | False | 348 (21.3) | 719 (78.8) |
| *Follow-up tests* | Chorionic villus sampling and amniocentesis can demonstrate with more certainty than NIPT whether there is a trisomy in the fetus. | True | 1,484 (90.8) | 885 (96.9) |
| | If the first trimester combined test shows an increased risk of 1 in 200 or higher, the woman can then still opt for NIPT. | True | 1,521 (93.0) | 826 (90.5) |
| | If the nuchal fold measurement in the first trimester combined test is ≥3.5 mm, and the karyotyping appears normal in the subsequent invasive test, the parents can be reassured. | False | 1,081 (66.1) | 633 (69.3) |
| *Fetal structural Anomaly Scan* | A Fetal Anatomy Scan is used to investigate physical abnormalities in an unborn baby. | True | 1,613 (98.7) | 901 (98.7) |
| | The primary responsibility for making a medical indication for Advanced Ultrasound Examination lies with the counselor. | True | 1,498 (91.6) | 840 (92.0) |

NIPT = Non-Invasive Prenatal Test, FCT = First trimester Combined Test. Grey fields indicate that <70% of the counselors answered an item correctly.

## Attitudes regarding the blended learning program

At T1, most counselors had a positive attitude towards the mandatory blended learning program for counselors; 648 (74%) valued the blended learning as 'good', 130 (14.9%) as 'not good', and 97 (11.1%) counselors answered 'other'. Examples of remarks were: 'once every two years a blended learning training is too often', 'okay, if online education is used where appropriate', 'too much regarding counseling skills training', and 'okay, only if major changes in (medical) developments have been made'.

**Table 3. Overall mean scores of counselors' knowledge represented per prenatal screening region.**

| Region[1] | Overall mean knowledge scores at T0 | Overall mean knowledge scores at T1 |
|---|---|---|
| | M (SD) | M (SD) |
| 1 | 24.3 (3.9) | 29.6 (2.7) |
| 2 | 22.5 (3.7) | 28.9 (2.8) |
| 3 | 22.6 (3.5) | 28.7 (2.8) |
| 4 | 22.9 (3.5) | 28.0 (4.3) |
| 5 | 23.4 (3.7) | 28.8 (3.1) |
| 6 | 23.4 (3.7) | 28.6 (2.8) |
| 7 | 24.0 (4.0) | 28.2 (2.9) |

[1] Data of the two Regional Centers for prenatal screening in Amsterdam are merged.

## Discussion

Our study aimed to assess the results of a mandatory blended learning program on prenatal screening counselors' knowledge as part of the Dutch national implementation of NIPT. The learning program was designed to bridge knowledge gaps identified at the time NIPT was transitioned from a second-tier test, for women at an increased risk for common trisomies based on the FCT or medical history, to a first-tier test available to all pregnant women. The program was intended to be a flexible and effective framework incorporating a variety of media and methods such as e-learning and face-to-face seminars. The second voluntary knowledge assessment took place approximately nine months after NIPT transition and followed administration of the blended learning program.

Counselors, the majority of whom were midwives (80%), answered more knowledge items correctly after NIPT implementation and blended learning (76% versus 66%). Knowledge levels were positively associated with years of work experience and a secondary care work-setting. The majority (75%) had positive attitudes towards NIPT at the time of the first assessment; this rose significantly to 85% in the second survey. About three-fourth of the participating counselors had a positive attitude towards the mandatory blended learning program.

In line with previous studies in the Netherlands, counselors' knowledge was generally considered sufficient [6]. Although counselors' knowledge overall improved after NIPT education and several months of experience with NIPT as a first-tier test, specific areas still showed a need for improvement, despite the tailored education provided. Knowledge on the theme *Trisomy 13, 18 and 21* did not improve over time; only three out of six questions were answered correctly by >70% at both T0 and T1. The incorrectly answered questions concerned development delays in Down syndrome, the fact that most children with Down syndrome are born to women younger than 36 years old, and the survival rates of a child with trisomy 13. The theme *test characteristics of NIPT* showed significant improvement, with 2/6 questions answered correctly by >70% at T0 increasing to 5/6 questions answered correctly at T1. The question most counselors struggled with in this theme was the false positive rate of Trisomy 13. Still, the percentage of respondents who correctly answered this question rose from 11% to 47%. The theme *additional findings of NIPT and FCT* also showed significant improvement in rates of correctly answered questions. The items about monochorionic pregnancies in the theme *In- and exclusion-criteria NIPT and FCT* were still only answered correctly by 53–63% of the counselors, at T1. This could be explained by the fact that most of the counselor work in a primary care setting and did not counsel monochorionic pregnancies, who were mostly counselled in hospitals.

The finding that not all post-implementation questions were answered correctly by >70% of the counselors may be due to the fact that some questions contained detailed knowledge of rare conditions and precise numbers which might be hard to remember, given the lack of daily use of such information. Nevertheless, optimisation of knowledge is essential since women rely on information provided by counselors to make an informed decision about prenatal aneuploidy screening, and can also influence their perception of the quality of life for children with anomalies included in NIPT. The need for providing more information on living with anomalies such as Down syndrome has been emphasised [31].

From the literature, we know that the baseline level of a counselors' knowledge is associated with their professional background, and ongoing participation in educational programs for counselors is effective in improving knowledge levels [6, 13]. A Dutch survey study among primary care midwives showed that a positive attitude regarding prenatal screening for Down syndrome was positively associated with knowledge levels [32]. Although we observed a rise in the percentage of counselors who showed positive attitudes towards NIPT, we found no

association between these attitudes and counselors' knowledge. One explanation might be that the majority of counselors already had positive attitudes. In addition, although the uptake of NIPT differs substantially between regions (ranging from 32% to 68%) [33], according to our results, the level of counselors' knowledge did not.

Consistent with other research [5], our respondents valued participating in educational programming regarding prenatal aneuploidy screening to improve their skills and stay up-to-date. However, also in line with other research [34], some counselors commented that the blended learning program should add something to their competencies, which might be independent of the frequency of once every two years. This outcome was also acknowledged by Dutch policymakers, resulting in a thorough evaluation of the blended learning program in 2019, and the ongoing redesign of content and program frequency in 2021 [35].

A strength of our study was the relatively high response rate. Furthermore, we recruited counselors from all over the country, resulting in a study population that was representative of the Dutch counselor population. However, we could not link the individual answers of participants in the first questionnaire to those who participated in the second one. Therefore, we do not know how much counselors did participate at both measurements. Moreover, the sole impact of the blended learning program cannot be distinguished from the nine months of 'learning by doing' through the implementation of first-tier NIPT. Nevertheless, our results suggest that knowledge levels in most topics were significantly improved after nine months, which by itself is promising and maybe a starting point for future research into combining a blended learning educational program with clinical practice. Finally, at T1 the counselors' mean years of working experience was one year less compared to participants at T0, which might have led to an underestimation of knowledge levels during the second assessment, since knowledge was positively associated with the amount of work experience.

Regarding practical implications, policy makers and educators could use our study results to reflect upon the body of knowledge that counselors for prenatal aneuploidy screening should have. An interdisciplinary panel of experts could help develop such body of knowledge; through interdisciplinary dialogue, counselors can learn about the effect of their counseling on clients views and questions raised once they received a positive screening result. Furthermore, our results stress the importance of assessing knowledge levels of counselors through questionnaires that contain essential and unambiguous content, with a focus on knowledge that all counselors should have. Therefore, the development and use of diagnostic tools to accurately measure counselors' knowledge may help to increase the awareness of personal knowledge gaps and the formulation of individual learning goals, before participating in educational training programs.

## Conclusions

A mandatory blended learning program alongside the implementation of NIPT as a first-tier screening test in the Dutch prenatal screening program significantly improved counselors' knowledge about prenatal aneuploidy screening. However, certain knowledge areas showed no or little improvement, revealing that educational tools need to be better tailored to identify and bridge the knowledge gaps for prenatal aneuploidy counselors. Individualized learning goals might optimize the effectiveness of the NIPT blended learning program. More research is needed to evaluate the effect of the blended learning program on higher levels of learning such as 'knows how', 'shows how' and 'does', given the essential role counselors have to comprehensively inform pregnant women about prenatal testing.

## Supporting information

**S1 Appendix. Dutch prenatal screening setting.**
(DOCX)

**S2 Appendix. Development of the blended learning program.**
(DOCX)

**S3 Appendix. Members of the Dutch NIPT consortium.**
(PDF)

## Acknowledgments

We gratefully acknowledge the contribution of counselors who completed the questionnaires for this study and the Regional Centres for prenatal screening for their help in recruitment. We further acknowledge Daphne de Jong and Caja Schouten for assisting in the study at the time of their Bachelor Midwifery.

## Author Contributions

**Conceptualization:** Linda Martin, Janneke T. Gitsels-van der Wal, Caroline J. Bax, Lidewij Henneman.

**Data curation:** Linda Martin.

**Formal analysis:** Linda Martin.

**Methodology:** Linda Martin.

**Project administration:** Linda Martin.

**Supervision:** Lidewij Henneman.

**Writing – original draft:** Linda Martin.

**Writing – review & editing:** Linda Martin, Janneke T. Gitsels-van der Wal, Caroline J. Bax, Mijntje J. Pieters, Jacqueline C. I. Y. Reijerink-Verheij, Robert-Jan Galjaard, Lidewij Henneman.

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
