## [Decision Letter · Decision Letter 0]

9 Feb 2022

PONE-D-22-02421Concerning Submission of Research Nationwide implementation of the Non-Invasive Prenatal Test: Evaluation of a blended learning program for counselorsPLOS ONE

Dear Dr. Martin,

Thank you for submitting your manuscript to PLOS ONE. After careful consideration, we feel that it has merit but does not fully meet PLOS ONE’s publication criteria as it currently stands. Therefore, we invite you to submit a revised version of the manuscript that addresses the points raised during the review process.

We look forward to receiving your revised manuscript.

Kind regards,

Antonio Simone Laganà, M.D., Ph.D.

Academic Editor

PLOS ONE

Journal Requirements:

4. Please ensure that you refer to Figure 1 in your text as, if accepted, production will need this reference to link the reader to the figure.

Additional Editor Comments:

The topic of the manuscript is interesting. Nevertheless, the reviewers raised several concerns: considering this point, I invite authors to perform the required major revisions.

Reviewers' comments:

Reviewer's Responses to Questions

**Comments to the Author**

1. Is the manuscript technically sound, and do the data support the conclusions?

Reviewer #1: Yes

Reviewer #2: Yes

2. Has the statistical analysis been performed appropriately and rigorously? 

Reviewer #1: Yes

Reviewer #2: No

3. Have the authors made all data underlying the findings in their manuscript fully available?

Reviewer #1: No

Reviewer #2: No

4. Is the manuscript presented in an intelligible fashion and written in standard English?

Reviewer #1: Yes

Reviewer #2: Yes

5. Review Comments to the Author

Reviewer #1: Dear Authors, I found your paper interesting. I believe strong effort should be put in the implementation of campaign to acknowledge health care professionals regarding NIPT.

This test has been a large diffusion, but, women do not perform an adequately informed choice.

Your paper highlight the importance of programs to improve de adequacy of health care professionals counseling regarding NIPT, therefore good job!

I have some minor revisions:

1) I would try to ameliorate the abstract readability, is a bit tortuous, abstract is the first insight into a paper therefore describe your job in an easier way.

- I would have talk about prenatal aneuploidy screening not anomaly screening

- I would have specified who do you mena by Duch prenatal counselors (obstetricians, midwives, family doctors

- Sentence linke " were positively associated with more correctly answered knowledge questions"

I would have said positively associated with a higher level of knowledge for example

In the mean text

The introduction

1) Again discuss about prenatal screening for aneuploidies not anomalies

2) line 62 is essential to facilitate informed choice for expectant parents regarding prenatal

63 screening" I would have said to allow an effective/adequate pregnant women's informed choice

line 89

I would suggest to mention this paper PMID: 33111167, that again shows how an not-adequate counseling from "counselor" associate with a poor understanding of women regarding NIPT, often considered as "diagnostic" and not "screening"

I would recommend your paper to be published with minor revisions

Reviewer #2: This manuscript evaluations the impact of a training program. These kinds of evaluations are useful to inform other providers of similar educational initiatives. The manuscript can be improved by addressing the following issues:

1. Introduction: Provide the reader with a brief description of what Non-Invasive Prenatal Testing is.

2. Introduction: What are first and second-tiers? Anyone from a different health system does not know what this means.

3. Introduction: line 75 - I would not use the word 'unacceptable' - replace it with something like 'problematic'

4. Introduction: line 80 - what is interfering education by commercial test providers?

5. Introduction: line 86-87 It is not correct to say that little is know about how efficient and effective educational programs are for healthcare implementations. There is plenty of literature about education on other topics.

6. Materials and Methods: line 134 - what is RIVM-CvB and RC (spell out acronyms at first use)

7. Materials and Methods line 176 - an unpaired t-test is not the correct statistical test. The sample is presumably mostly the same participants at both assessments. I understand that they can't be matched, but the mean score on the first assessment could be used as the threshold score for the comparison of the posttest scores using a one-sample t test.

8. Materials and Methods line 178-179: Why are separate regressions used for the pre and post?

6. PLOS authors have the option to publish the peer review history of their article (what does this mean?). If published, this will include your full peer review and any attached files.

Reviewer #1: No

Reviewer #2: No

---

## [Author Response · Author response to Decision Letter 0]

1 Mar 2022

March, 1st 2022 

Dear reviewers,

Thank you for your expert review of our manuscript titled “Nationwide implementation of the Non-Invasive Prenatal Test: Evaluation of a blended learning program for counselors”, PONE-D-22-02421.

We have used your comments to improve our paper further, and we are pleased to send you our revised manuscript. Our response after each comment in italics are provided below.

Reviewer #1: 

Dear Authors, I found your paper interesting. I believe strong effort should be put in the implementation of campaign to acknowledge health care professionals regarding NIPT.

This test has been a large diffusion, but, women do not perform an adequately informed choice.

Your paper highlight the importance of programs to improve the adequacy of health care professionals counseling regarding NIPT, therefore good job!

Thank you for your encouraging words!

I have some minor revisions:

1) I would try to ameliorate the abstract readability, is a bit tortuous, abstract is the first insight into a paper therefore describe your job in an easier way. 

We do agree with the reviewer that the abstract could be made more readable. Therefore, we changed is according to the suggestions below (line 29-43 of the manuscript): “This study assesses the results of a mandatory blended learning-program for counselors (e.g. midwives, sonographers, obstetricians) guiding national implementation of the Non-Invasive Prenatal Test (NIPT). We assessed counselors' 1) knowledge about prenatal aneuploidy screening, 2) factors associated with their knowledge (e.g. counselors' characteristics, attitudes towards NIPT), and 3) counselors’ attitudes regarding the blended learning.”

- I would have talk about prenatal aneuploidy screening not anomaly screening

We do understand this suggestion. Since we measured not only knowledge about aneuploidy screening but also about structural anomaly screening we used the term prenatal anomaly screening. However, most knowledge items concerned aneuploidy screening and the article is about implementation of NIPT. Therefore, we changed our wording into prenatal aneuploidy screening (line 32; see inserted text comment 1).

- I would have specified who do you mean by Duch prenatal counselors (obstetricians, midwives, family doctors).

We implemented this suggestion (line 30): “This study assesses the results of a mandatory blended learning-program for counselors (e.g. midwives, sonographers, obstetricians) guiding national implementation of the Non-Invasive Prenatal Test (NIPT). We assessed counselors' 1) knowledge about prenatal aneuploidy screening, 2) factors associated with their knowledge (e.g. counselors' characteristics, attitudes towards NIPT), and 3) counselors’ attitudes regarding the blended learning.”

- Sentence linke " were positively associated with more correctly answered knowledge questions". I would have said positively associated with a higher level of knowledge for example.

Thank you for this more readable suggestion (see line 43): “Work experience and secondary care work-setting were positively associated with a higher level of knowledge.”

In the mean text

The introduction

1) Again discuss about prenatal screening for aneuploidies not anomalies.

In line with our response to the earlier comment on this topic we changed our wording into prenatal aneuploidy screening thoughout the manuscript.

2) line 62 is essential to facilitate informed choice for expectant parents regarding prenatal

63 screening" I would have said to allow an effective/adequate pregnant women's informed choice

We agree with the reviewer that the suggested text improves the manuscript. We therefore edited the text into: “Such knowledge, e.g. understanding the advantages and limitations of NIPT compared to other prenatal tests and knowing how to interpret the results, is essential to enable pregnant women making an adequate informed choice regarding prenatal screening (line 63).”

3) line 89

I would suggest to mention this paper PMID: 33111167, that again shows how an not-adequate counseling from "counselor" associate with a poor understanding of women regarding NIPT, often considered as "diagnostic" and not "screening"

Thank you for putting our attention towards this interesting article. We now refer to this article at line 75-78: “The resulting knowledge gaps are problematic since insight into NIPT's test characteristics is fundamental for understanding the clinical implications of the test results and the fact that NIPT is not diagnostic, diagnostic options and thereby for providing adequate counseling that supports informed decisions [4,6,14,35].”

Reviewer #2: This manuscript evaluations the impact of a training program. These kinds of evaluations are useful to inform other providers of similar educational initiatives. The manuscript can be improved by addressing the following issues:

1. Introduction: Provide the reader with a brief description of what Non-Invasive Prenatal Testing is.

We thank the reviewer for this relevant suggestion. In lines 55-57 we briefly describe NIPT: 

“NIPT is a maternal blood test using cell-free DNA, mostly targeted at the common aneuploidies Down-, Edwards- and Patau syndrome. In most countries, NIPT is readily commercially available, although it has increasingly been offered as a second-tier pr first-tier an aneuploidy screening test, within nationwide healthcare programs, in which mostly primary maternity care providers offer accompanying pre-test counseling [1-3].”

2. Introduction: What are first and second-tiers? Anyone from a different health system does not know what this means.

NIPT can be offered as a first-tier screening test to all pregnant women regardless of their risk level, or as a second-tier test to women with increased risk for fetal aneuploidies only. This has been added to the introduction (line 97-102): 

“In 2017, alongside the transition of NIPT from a second-tier test, only available for women with a high risk pregnancy based on the First trimester Combined Test results or medical indication, to a first-tier screening test for all pregnant women, the Dutch government set new educational requirements for healthcare professionals who provide counseling for prenatal aneuploidy screening [18,19]” 

3. Introduction: line 75 - I would not use the word 'unacceptable' - replace it with something like 'problematic'

We replaced the word ‘unacceptable’ for ‘problematic’, according to the reviewer’s suggestion (line 75-76).

4. Introduction: line 80 - what is interfering education by commercial test providers?

By intefering education we ment biased education and changed the text accordingly (see line 81).

5. Introduction: line 86-87 It is not correct to say that little is know about how efficient and effective educational programs are for healthcare implementations. There is plenty of literature about education on other topics.

We agree with the reviewer that line 86-87 is not correct. We corrected into (line 89): 

“Furthermore, relatively little is known about efficient and effective educational programs accompanying implementation of NIPT, ….etc.”

6. Materials and Methods: line 134 - what is RIVM-CvB and RC (spell out acronyms at first use)

In line 141, in the revised manuscript, we spell out the acronym RIVM-CvB (Centre for Population Screening at the National Institute for Public Health and the Environment). In 123 we already spell out RC (Regionals Centers for prenatal screening). 

7. Materials and Methods line 176 - an unpaired t-test is not the correct statistical test. The sample is presumably mostly the same participants at both assessments. I understand that they can't be matched, but the mean score on the first assessment could be used as the threshold score for the comparison of the posttest scores using a one-sample t test.

Thank you for this remark! In line with the reviewers comment we did discuss which statistical test we should use. By reading the manuscript again we do see a discrepance in our work. In line 185 we wrote: Mean scores at T0 and T1 were statistically evaluated using an unpaired sample t-test. However, in line 194 we wrote: One sample proportion tests were performed to identify significant differences between the results of T0 and T1. 

We did the latter, but unfortunately we did not remove line 185 from the manuscript. Now we did.

8. Materials and Methods line 178-179: Why are separate regressions used for the pre and post?

By using separate models for T0 and T1 we took a conservative approach, because we did not want to lean on assumptions about overlapping groups within T0 and T1. To clarify this to the readers we edited the text (line 188-190) into:

“Separate models were analyzed for T0 and T1, because we did not want to lean on the assumption that the same participants were included at both assessments.”

---

## [Decision Letter · Decision Letter 1]

18 Apr 2022

Concerning Submission of Research Nationwide implementation of the Non-Invasive Prenatal Test: Evaluation of a blended learning program for counselors

PONE-D-22-02421R1

Dear Dr. Martin,

We’re pleased to inform you that your manuscript has been judged scientifically suitable for publication and will be formally accepted for publication once it meets all outstanding technical requirements.

Kind regards,

Antonio Simone Laganà, M.D., Ph.D.

Academic Editor

PLOS ONE

Additional Editor Comments (optional):

I carefully evaluated the revised version of this manuscript.

Authors have performed the required changes, improving significantly the quality of the paper.

Reviewers' comments:

Reviewer's Responses to Questions

**Comments to the Author**

1. If the authors have adequately addressed your comments raised in a previous round of review and you feel that this manuscript is now acceptable for publication, you may indicate that here to bypass the “Comments to the Author” section, enter your conflict of interest statement in the “Confidential to Editor” section, and submit your "Accept" recommendation.

Reviewer #1: All comments have been addressed

Reviewer #2: All comments have been addressed

2. Is the manuscript technically sound, and do the data support the conclusions?

Reviewer #1: Yes

Reviewer #2: Yes

3. Has the statistical analysis been performed appropriately and rigorously? 

Reviewer #1: Yes

Reviewer #2: Yes

4. Have the authors made all data underlying the findings in their manuscript fully available?

Reviewer #1: Yes

Reviewer #2: (No Response)

5. Is the manuscript presented in an intelligible fashion and written in standard English?

Reviewer #1: Yes

Reviewer #2: Yes

6. Review Comments to the Author

Reviewer #1: Well done, I think the paper has been improved. The topic as I've already commented is actual and interesting therefore

i will recommend it for publication

Reviewer #2: The authors have done a good job of responding to the review feedback. I have no further suggestions.

7. PLOS authors have the option to publish the peer review history of their article (what does this mean?). If published, this will include your full peer review and any attached files.

Reviewer #1: No

Reviewer #2: No

---

## [Editor Report · Acceptance letter]

22 Apr 2022

PONE-D-22-02421R1 

Nationwide implementation of the Non-Invasive Prenatal Test: Evaluation of a blended learning program for counselors 

Dear Dr. Martin:

I'm pleased to inform you that your manuscript has been deemed suitable for publication in PLOS ONE. Congratulations! Your manuscript is now with our production department. 

Kind regards, 

on behalf of

Dr. Antonio Simone Laganà 

Academic Editor

PLOS ONE